

# Flower diversity and bee reproduction in an arid ecosystem

Jimena Dorado[1] and Diego P. Vázquez[1,2]

[1] Instituto Argentino de Investigaciones de Zonas Áridas, Consejo Nacional de Investigaciones Científicas y Técnicas, Mendoza, Argentina
[2] Facultad de Ciencias Exactas y Naturales, Universidad Nacional de Cuyo, Mendoza, Argentina

## ABSTRACT

**Background:** Diverse flower communities are more stable in floral resource production along the flowering season, but the question about how the diversity and stability of resources affect pollinator reproduction remains open. High plant diversity could favor short foraging trips, which in turn would enhance bee fitness. In addition to plant diversity, greater temporal stability of floral resources in diverse communities could favor pollinator fitness because such communities are likely to occupy the phenological space more broadly, increasing floral availability for pollinators throughout the season. In addition, this potential effect of flower diversity on bee reproduction could be stronger for generalist pollinators because they can use a broader floral spectrum. Based on above arguments we predicted that pollinator reproduction would be positively correlated to flower diversity, and to temporal stability in flower production, and that this relationship would be stronger for the most generalized pollinator species.

**Materials and Methods:** Using structural equation models, we evaluated the effect of these variables and other ecological factors on three estimates of bee reproduction (average number of brood cells per nest per site, total number of brood cells per site, and total number of nests per site), and whether such effects were modulated by bee generalization on floral resources.

**Results:** Contrary to our expectations, flower diversity had no effect on bee reproduction, stability in flower production had a weakly negative effect on one of the bee reproductive variables, and the strength of the fitness-diversity relationship was unrelated to bee generalization. In contrast, elevation had a negative effect on bee reproduction, despite the narrow elevation range encompassed by our sites.

**Discussion:** Flower diversity did not affect the reproduction of the solitary bees studied here. This result could stem from the context dependence of the diversity-stability relationship, given that elevation had a positive effect on flower diversity but a negative effect on bee reproduction. Although high temporal stability in flower production is expected to enhance pollinator reproduction, in our study it had a weakly negative—instead of positive—effect on the average number of brood cells per nest. Other environmental factors that vary with elevation could influence bee reproduction. Our study focused on a small group of closely-related bee species, which cautions against generalization of our findings to other groups of pollinators. More studies are clearly needed to assess the extent to which pollinator demography is influenced by the diversity of floral resources.

Corresponding author
Jimena Dorado,
jdorado@mendoza-conicet.gob.ar

## INTRODUCTION

There is a consensus that diversity enhances ecosystem functioning (*Cardinale et al., 2012*). Species diversity provides redundancy in function so that ecological processes are more stable in more diverse communities (*MacArthur, 1955*; *Elton, 1958*). In plant communities, the diversity–stability relationship has been well studied for biomass production (*Caldeira et al., 2005*; *Tilman, Reich & Knops, 2006*; *Isbell, Polley & Willsey, 2009*; *Hector et al., 2010*), and we have recently reported that diverse flower communities are also more temporally stable in terms of floral resource production (*Dorado & Vázquez, 2014*). However, the question about how the diversity and stability of resources affect reproduction of pollinators remains open.

It is well known that ecosystem productivity is positively associated to species diversity (*Cardinale et al., 2012*). We propose that a similar effect of plant species diversity can be expected on population- and community-level productivity of pollinators (i.e., reproductive output or biomass), for several reasons. First, the probability that a resource species important for reproduction is present increases with species diversity (the "sampling effect"; *Loreau, 2010*). Second, greater plant diversity can lead to reduced foraging trip duration (e.g., *Gathmann & Tscharntke, 2002*), which could mean more energy available for reproduction (*Minckley et al., 1994*; *Zurbuchen et al., 2010*). Third, if different plant species in the community offer complementary resources (e.g., they cover non-overlapping nutritional needs of pollinators), greater plant diversity could mean a greater probability of meeting the nutritional needs of pollinators (see, e.g., *Williams & Tepedino, 2003*). The effect of flower diversity on bee reproduction should be stronger for polylectic than oligolectic pollinators, given that the latter are more restricted in their diet.

In addition to plant diversity, greater temporal stability of floral resources in diverse communities (*Dorado & Vázquez, 2014*) could favor pollinator fitness because such communities are likely to occupy the phenological space more broadly than their species-poor counterparts, increasing floral availability for pollinators throughout the season. For example, in multi-species assemblages of herbaceous plants of the genus *Clarkia*, diverse communities provide more resources along the flowering season, sustaining a higher number of pollinator individuals per plant (*Moeller, 2004*). Furthermore, a bumblebee study found that even if floral resources are abundant, high stability of floral resources throughout the flowering season is needed to enhance bumblebee fitness (*Westphal, Steffan-Dewenter & Tscharntke, 2009*; *Rundlöf et al., 2014*). Thus, both high flower abundance and high temporal stability of floral resources are likely to enhance pollinator reproduction (*Müller et al., 2006*; *Westphal, Steffan-Dewenter & Tscharntke, 2009*).

To evaluate whether there is an effect of flower diversity on pollinator reproduction it is necessary to disentangle the effect of flower abundance, as it could be positively correlated with flower richness, as it happens with biomass in plant communities (*Tilman, 1999*); if so, there could be a spurious positive correlation between flower richness and pollinator fitness. Other local environmental factors, such as elevation or disturbance

history, should also be accounted for, as they are known to influence species diversity (*Potts et al., 2003*; *Grytnes & McCain, 2007*; *Dorado & Vázquez, 2014*). Structural equation modeling (SEM) represents an excellent tool to assess causal relationships among multiple variables simultaneously (*Grace, 2006*), as is the case in the present study.

Our aim is to study the effect of flower diversity and temporal stability of floral resources on the reproduction of a cavity nesting bee assemblage from the Monte desert in Argentina. Based on the above arguments, we expected to find that flower diversity and temporal stability of floral resources correlates positively to three estimates of bee reproduction at the population and community levels: average number of brood cells per nest per site, total number of brood cells per site, and total number of nests per site. We also expected to find a positive correlation between the strength of the reproductive output-diversity correlation and the degree of generalization of each bee species.

## METHODS

### Study area and sampling

This study was conducted in the Monte desert in Villavicencio Nature Reserve, located ca. 40 km north of Mendoza city, Argentina, during the 2008 flowering season (15 October–8 December 2008; authorized by Dirección de Recursos Naturales Renovables de la Provincia de Mendoza, approval numbers 1130 and 646). We worked in fourteen $100 \times 200$ m rectangular study sites (minimum and maximum distance between them were 1.11 and 14.13 km, respectively). These sites lie at 1,100–1,500 m above sea level, at the ecotone between the Monte desert and the Prepuna biomes (*Ambrosetti, Del Vitto & Roig, 1986*). The plant community is a 2 m tall shrubland dominated by *Larrea divaricata* (Zygophyllaceae), *Zuccagnia punctata* (Fabaceae), *Prosopis flexuosa* (Fabaceae), *Condalia microphylla* (Rhamnaceae), *Acantholippia seriphioides* (Verbenaceae), and *Opuntia sulphurea* (Cactaceae). We selected sites with different flower abundance, composition and diversity. The region suffers from recurrent fires, which are mostly human-caused and are in fact the most common human disturbance (E. L. Stevani, 2008, personal communication); the time elapsed since the last fire varied substantially among our study sites (Table S1).

### Trap nest sampling

We placed trap nests in six points per plot as shown in Fig. S1. Each point had two groups of 24 trap nests consisting of a wood piece with a longitudinal cavity of 5, 8 or 11 mm in diameter, and 15 cm of length for the smallest two diameters and 28 cm of length for the largest diameter; wood pieces were arranged as shown in Fig. S2. Trap nests were checked weekly; occupied traps were taken to the laboratory and replaced by empty ones. Each trap nest constitutes one bee nest. Once in the laboratory, nests were opened to record the number of cells; whenever the nest had more than one brood cell, one of them was extracted for pollen identification for other analyses (*Dorado et al., 2011*; *Vázquez et al., 2012*), and the rest was kept until adult emergence. The number of emerged adults and their taxonomic identity were recorded in all nests. Although adult number may be a good estimator of female fitness in the absence of larval mortality, because of the high rate

of nest parasitism recorded in our study we used instead the number of brood cells per nest as an estimate of female fitness. One species, *Trichothurgus laticeps* Friese, lacks brood cells, as females lay eggs bare amidst a pollen mass; thus, for this species we used the length of the trap cavity occupied by pollen as an estimate of the number of brood cells. For the analysis, we used only the seven bee species that occupied at least 30 trap nests, as we judged smaller sample sizes unreliable for statistical analyses.

## Plant sampling

Floral resource availability was studied using flower density, as flowers represent the resource packages encountered by pollinators as they forage (see also *Vázquez, Chacoff & Gagnolo, 2009*). Flower density was measured weekly at four 8 × 20 m plots and two 2 × 50 m transects per site, as described in Fig. S1. We considered weekly sampling intervals adequate, as flowers in our system usually last less than a week. Flower density was estimated multiplying the mean number of flowers per individual by the total number of flowering individuals in the transect or plot when individuals could be distinguished (shrubs and some herbs); we estimated the number of flowers per individual in at least ten individuals of the site. When it was not possible to identify flowering individuals (some herbaceous species), all flowers in a plot or transect were counted. We included in the study all flowering plant species that were assumed to be animal pollinated (we excluded only grass species).

## Statistical analysis

To evaluate the effects of flower diversity and temporal stability of floral resources on bee fitness and to assess the influence of other ecological factors on this relationship, we used structural equation models (hereafter SEM). We built a general initial model to explore the effects of flower richness, flower abundance, time elapsed since the last fire, elevation, and temporal stability in flower production on bee reproductive parameters. We estimated bee productivity at the community level using two proxies: total number of brood cells per site, and total number of nests per site. To evaluate reproduction at the species level we used three proxies: average number of brood cells per nest, total number of brood cells per site, and total number of nests per site. To estimate the average number of brood cells per nest we used only data of sites where species were present; for the total number of brood cells per site and the total number of nests per site we used data of all sites, as the absence of a species in a site represented zero abundance. Flower richness was used as a proxy of flower diversity; it was rarefied to remove the effect of flower abundance. Flower abundance was estimated as flower density per site. Time elapsed since the last fire was provided by park rangers (E. L. Stevani, personal communication). Temporal stability in flower production along the season was calculated as the inverse of its coefficient of variation (see *Dorado & Vázquez, 2014*). For each bee species, from the initial model we generated more parsimonious nested models by removing variables with small non-significant path coefficients (see models in Fig. 1; see also *Maestre et al., 2010*).

We evaluated alternative SEM models using a d-separation test (*Shipley, 2000*; *Shipley, 2013*). This analysis allowed us to select the best fitting model based on Akaike's

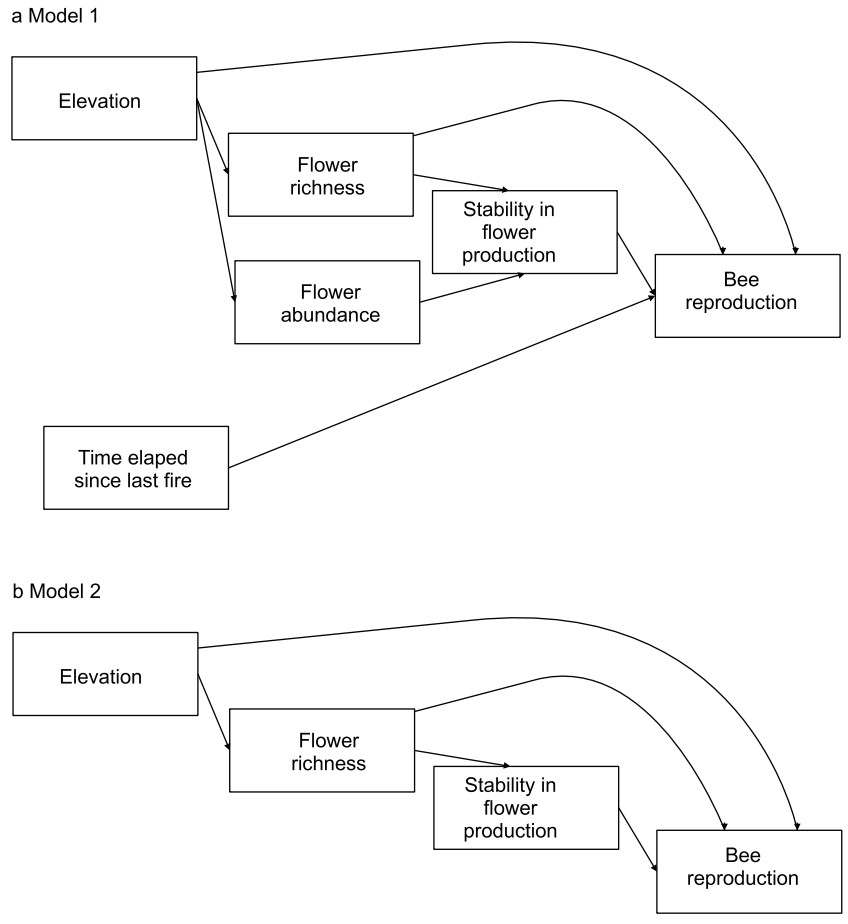

**Figure 1 Models evaluating the effect of flower diversity and other ecological factors on the reproductive variables of different bee species of the 14 study sites.** (A) Complete model. (B) Nested models generated by removing variables with non-significant effects or small path coefficients that were non-significant. Model 2 was selected by ΔAIC for all bee species.

information criterion (AIC) using a small sampling size. The d-separation test involves calculating a probability of independence, $p_i$, between two pairs of variables that are not directly connected with an arrow in the causal model, and then using those probabilities to calculate Fisher's $C$ statistic, which follows a chi-square distribution, $C = -2 \sum_{i=1}^{K} (\ln p_i)$ (*Shipley, 2000*; *Shipley, 2013*). The group of all $k$ pairs of independent variables with their corresponding conditional variables constitutes the basis set (Table S2). Independence probability should be estimated using an appropriate test; in our case we used the $p$-value associated to Pearson's partial correlation coefficient as an estimate of $p_i$. We then calculated the maximum likelihood estimate for each model using the $C$ value associated to each causal model, and the corrected Akaike's Information Criterion as AIC = 2 ln $C$ + 2$K$, where $K$ is the total number of free parameters in the model and $n$ is the sample size. To discriminate among competing models, we used the AIC difference, ΔAIC, between a given model and the best-fitting one, i.e., that with the lowest value of AIC. When ΔAIC < 3, models are generally considered to have substantial support; for 3 > ΔAIC < 7, models are considered to have considerably less

support, while for $\Delta$AIC > 10, models have essentially no support relative to the best model of the set (*Richards, 2005*; *Burnham & Anderson, 2010*). We used meta-analytical methods to evaluate whether the studied effects were general for all bee species. To apply the meta-analytical methods, the path coefficients from the SEM models for each bee species were normalized by applying Fisher's *z* transform, $z = 0.5 \ln [(1 + r)/(1 - r)]$ (*Zar, 1999*) to make them comparable. To weigh the correlation coefficients, we divided them by the inverse of the sampling variance, $w = 1/\text{var}(r) = N - 3$ (*Rosenthal, 1991*; *Zar, 1999*; *Gurevitch, Curtis & Jones, 2001*). We used a bootstrap resampling procedure written in R (*R Core Team, 2013*), with a sample size of 1,00,000, with which we calculated the mean and 95% percentile confidence limits of $z_w$ (*Manly, 1997*).

To evaluate whether the effect of flower diversity becomes stronger with increasing pollinator generalization, we performed Spearman's rank correlations between the path coefficient representing the effect of flower richness on each of the three bee reproductive parameters mentioned above, and two measures of the corresponding species degree of generalization. We estimated the degree of diet generalization of each bee species using the species degree and Simpson's diversity index; degree is simply the number of food species consumed from all sites polled, whereas Simpson's index is a function of the number of food items and the proportion in which they were consumed. We used rarefaction to estimate both measures of generalization to make them comparable among bee species, as the number of brood cells was highly variable among nests. A positive correlation between the path coefficient of flower richness on bee reproduction and generalization would support our hypothesis that the reproduction of generalist pollinators is enhanced by flower richness.

All analyses were performed using R statistical software (*R Core Team, 2013*). Rarefaction of flower richness was performed using the rarefy function of the vegan package (*Oksanen et al., 2013*). Pearson's partial correlations were performed using the pcor.test function of the ppcor package to obtain independence probabilities and the path coefficients (*Kim & Yi, 2007*; *Kim & Yi, 2006*; *Johnson & Dean, 2002*).

## RESULTS

We recorded 598 occupied trap nests by 11 solitary bee species (Table 1).

The complete model assessing the influence of multiple ecological factors on the potential relationship between flower diversity and bee reproduction at a community level (Model 1, Fig. 1A) showed a negative effect of elevation on bee reproductive variables, and no effect of the other evaluated factors on bee reproduction (Table 2).

The complete model assessing the influence of multiple ecological factors on the potential relationship between flower diversity and each bee species fitness (Model 1, Fig. 1A) showed no effect of flower abundance or time elapsed since the last fire on the bee reproductive variables studied. Although the only significant effect was that of temporal stability on the average number of brood cells per nest, we kept flower richness and elevation in the simplified model because they showed suggestive, albeit non-significant, trends. There was a weak positive effect of flower richness on average number of brood cells per nest and a weak negative effect of elevation on the three reproductive variables (Fig. S3);

**Table 1 Number of nests per species.** We used in this study the species that had more than 30 nests.

| Bee species | Occupied trap nests |
| --- | --- |
| *Anthidium andinum* Jörgensen | 6 |
| *Anthidium decaspilum* Moure | 54 |
| *Anthidium rubripes* Friese | 31 |
| *Anthidium vigintipunctatum* Friese | 39 |
| *Megachile* leucographa Friese | 222 |
| *Megachile* sp. C | 17 |
| *Megachile ctenophora* Holmberg | 74 |
| *Mourecotelles triciliatus* Toro & Cabezas | 3 |
| *Trichothurgus laticeps* Friese | 59 |
| *Xylocopa atamisquensis* Lucia & Abrahamovich | 88 |
| *Xylocopa splendidula* Lepertier | 5 |

**Table 2 Path coefficients of Models 1 for community bee reproduction.**

| Model | Variables | Path coefficients | *p*-value |
| --- | --- | --- | --- |
| 1 | Elevation → Flower richness | 0.39 | 0.15 |
| | Elevation → Flower abundance | −0.51 | 0.06 |
| | Flower richness → Stability | −0.54 | 0.04 |
| | Elevation → Total brood cells | −0.59 | 0.03 |
| | Elevation → Total nests | −0.57 | 0.04 |
| | Flower abundance → Total brood cells | 0.26 | 0.43 |
| | Flower abundance → Total nests | 0.24 | 0.48 |
| | Flower richness → Total brood cells | 0.27 | 0.80 |
| | Flower richness → Total nests | 0.38 | 0.23 |
| | Stability → Total brood cells | 0.35 | 0.35 |
| | Stability → Total nests | 0.30 | 0.35 |
| | Time elapsed since last fire → Total brood cells | 0.35 | 0.28 |
| | Time elapsed since last fire → Total nests | 0.45 | 0.14 |

however, none of these effects were statistically significant. The simplified model (Model 2, Fig. 1B) fitted best according to ΔAIC (8.43) for all species (see Figs. S1 and 1); this model fits the data well according to the d-separation test ($p = 0.84$, $df = 2$, $C = 3.69$). In this simplified model, the negative effect of temporal stability in flower production on the average number of cells per nest was weaker than in the complete model (Fig. 2 blue error bars; confidence limits of path coefficient for Model 1: −0.467, −0.101; confidence limits of path coefficient of Model 2: −0.264, −0.004). Also, the simplified model shows a negative trend in the effect of elevation on the total number of cells and nests per site, but this trend is statistically non-significant (confidence limits of path coefficients for the total number of cells and nests per site respectively: −0.465, 0.031, and −0.472, 0.027).

The effect of flower diversity on pollinator reproduction was unrelated to pollinator generalization for any of the bee reproductive variables and generalization indexes used (Table 3).
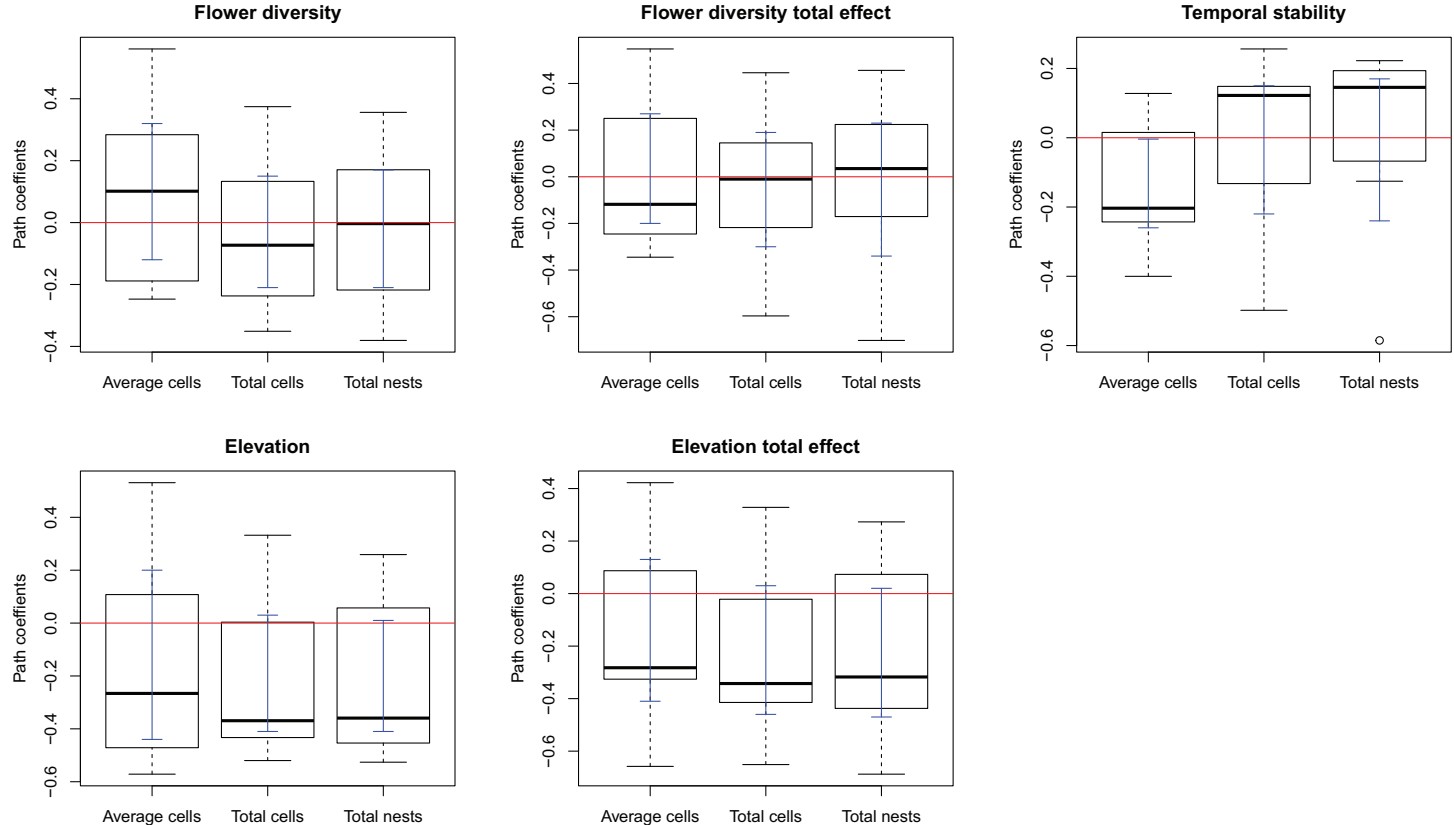

**Figure 2 Box-plot summarizing the path coefficients of Model 2 (see Fig. 1) for the seven bee species studied here.** In each box plot, the middle line indicates median, box limits are the first and third quartiles, whiskers indicate most extreme points $\leq$ 1.5 times the interquartile range, and circles indicate outliers of the seven path coefficients of the corresponding effect. Model 2 describes the effect of flower diversity (estimated using flower richness), temporal stability of flower production along the flowering season (estimated as the inverse of the coefficient of variation of the weekly flower abundance mean), and elevation (m above sea level) on three bee reproductive variables: "Average cells," the average number of brood cells per nest per site; "Total cells," the total number of brood cells per site; and "Total nests," the total number of nests per site. The ordinates represent the path coefficients; the abscissa represent the effect of the above ecological variables on bee reproductive variables. Blue error bars are the ninety-five percent confidence limits of path coefficients obtained from bootstrap sampling of the distribution of path coefficients.

**Table 3 Correlation coefficients between effect of flower diversity on pollinator reproduction and bee generalization.**

| Generalization index | Reproductive variable | Correlation coeficient ($r$) | $p$-value | N |
|---|---|---|---|---|
| Degree | Average number of cells per nest | −0.21 | 0.66 | 7 |
| Degree | Total number of cells per site | 0.42 | 0.35 | 7 |
| Degree | Total number of nests per site | 0.39 | 0.39 | 7 |
| Simpson's diversity index | Average number of cells per nest | 0.14 | 0.78 | 7 |
| Simpson's diversity index | Total number of cells per site | 0.46 | 0.30 | 7 |
| Simpson's diversity index | Total number of nests per site | 0.28 | 0.55 | 7 |

## DISCUSSION

Contrary to our expectations, we found no effects of flower diversity and flower abundance on bee reproduction, either at the community or at the species level. Thus, flower diversity did not matter for the reproduction of the solitary bees studied here.

Considering the ecosystem functioning context where relationships are commonly saturating (*Cardinale et al., 2012*), there is a possibility that we have sampled plant diversities corresponding only to the saturating part of the diversity-productivity curve. In addition, this result could stem from the context dependence of the diversity-stability relationship (*Griffin et al., 2010*), given that elevation had a positive effect on flower diversity (*Dorado & Vázquez, 2014*) but a negative effect on bee reproduction (Table 2). This trend in the effect of elevation on bee reproduction was observed despite the narrow elevation range encompassed by our sites (1,100–1,500 m), which suggests that the environmental conditions of the study sites could have influenced the relationship between floral diversity and bee reproduction.

An explanation of the negative effect of temporal stability on brood cell production concerns a compensatory behavior of females to avoid parasitism. In sites with high temporal stability in flower production, females might lay fewer eggs per nest while building more nests, so as to maximize larval survival per site. This reasoning makes two implicit assumptions. First, that the bee species are parasitized, which we indeed observed for many of the bee species studied here. Second, that nesting sites are not limited for the population. In fact, the trap nest sampling with replacement highly increased the nest availability in our study sites. If this mechanism were responsible for the observed negative effect of temporal stability in flower production on the average number of brood cells per nest, we would expect the number of cells per site to be either unrelated to temporal stability or to be higher in the more temporally stable sites, and the number of nests to be higher in the more temporally stable sites, as females would be laying eggs at their maximum capacity but distributing them in more nests. Matching these expectations, the total number of brood cells per site was unrelated to temporal stability (Fig. 2, flower stability), while the number of nests per site tends to increase with temporal stability for most species, although the effect was statistically non-significant.

An alternative explanation of the negative effect of temporal stability on brood cell production per nest could be that elevation might be weakening the effects of other variables on pollinator reproduction. This is particularly likely considering the positive direct effect of elevation on flower diversity, the positive indirect effect of elevation on stability, and the negative effect of elevation on bee reproduction (Table 2 and Fig. 2); these effects could be neutralizing the effect of the flower diversity and temporal stability in flower production on bee reproduction. The upper sites are located at the mouth of ravines, which are probably wetter and cooler than the lower sites, located in open land. Thus, changes in humidity and temperature associated to elevation could be influencing bee reproduction more strongly than the other ecological factors studied here.

We found no support for the idea that generalist bees are more favored in their reproduction by flower diversity than specialized ones, despite bee species included in this study having contrasting degrees of feeding specialization. Again, we think the negative effect of elevation on bee reproduction can be responsible for this unexpected result. It seems reasonable to think that species will respond idiosyncratically to flower

diversity and stability when there is context dependency, given our finding of no general effects of flower diversity on bee reproduction.

Although there is a consensus that diversity promotes ecosystem-level productivity (*Cardinale et al., 2012*), we failed to find this relationship at the community and population levels in our study. However, our study focused on a small group of closely-related bee species, representing less than 5% of the pollinator assemblage in our study area (*Chacoff et al., 2012*). Our study is in this sense limited, and our finding of no effects of floral diversity on pollinator demography cannot be generalized. More studies are clearly needed to assess the extent to which pollinator demography is influenced by the diversity of floral resources. These studies are becoming priority, as wild bees are known to enhance fruit production in crops, beyond the pollination service provided by honeybees (*Garibaldi et al., 2013*). Furthermore, although we have considered environmental factors, such as elevation or time post-fire, which appeared a priori good candidates to influence bee demography, other environmental factors may also be important. These include humidity and temperature, which should covary with elevation, and other biotic factors such as competition, predation and parasitism.

## ACKNOWLEDGEMENTS

We thank the administration of Villavicencio Natural Reserve for permission to conduct this study, the park rangers for help to find appropriated study sites in the field, Arturo Roig for help with bee identifications, Leticia Escudero, Nydia Vitale and Georgina Amico for laboratory assistance, and members of the Ecological Interactions Lab for helpful comments on the manuscript. JD is a postdoctoral fellow and DPV a career researcher with CONICET.

### Funding
Research was funded through grants from CONICET (PIP 6564), FONCYT (PICT 20805, 1471 and 2010-2779), and BBVA Foundation (BIOCON03-162). The funders had no role in study design, data collection and analysis, decision to publish, or preparation of the manuscript.

### Grant Disclosures
The following grant information was disclosed by the authors:
CONICET: PIP 6564.
FONCYT: PICT 20805, 1471 and 2010-2779.
BBVA: BIOCON03-162.

### Competing Interests
The authors declare that they have no competing interests.

## Author Contributions

- Jimena Dorado conceived and designed the study, conducted fieldwork, analyzed the data, and wrote the paper, prepared figures and/or tables.
- Diego P. Vázquez conducted fieldwork, and reviewed drafts of the paper, supervised the work for this article in his capacity as Ph.D. advisor.

## Field Study Permissions

The following information was supplied relating to field study approvals (i.e., approving body and any reference numbers):

Dirección de Recursos Naturales Renovables de la
Provincia de Mendoza 1130 and 646.

## Data Deposition

The raw data was supplied as Supplemental Dataset Files.

## Supplemental Information

Supplemental information for this article can be found online at http://dx.doi.org/10.7717/peerj.2250#supplemental-information.

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
