# Peer review of "Flower diversity and bee reproduction in an arid ecosystem"

_PeerJ, doi:10.7717/peerj.2250_

## Round 0.1 · original submission · Major Revisions

Please revise carefully the article addressing the requests of the three reviewers. In particular, kindly address the "Validity of the findings" comments from Reviewer 2 about replication.

Kind regards,

·

Basic reporting

- I'm not an English native speaker, but I feel English is adequate.
- Nice introduction, well embedded in the international context of related studies.
- Relevant prior literature well reported.
- Experimental hypothesis clearly indicated.
- Figures and table are well relevant to the general structure of the manuscript.

Experimental design

- Methods are clearly described, well appropriate and easily reproducible.
- Statistical approach is fine.
- Ethical standards respected

Validity of the findings

- Data are robust and interesting to scientific community.
- Conclusions are supported by findings.
- Discussion is nice and convincing.

Additional comments

I really enjoyed from reading the manuscript, nice paper.
I suggest only minor revisions:
- L22: please indicate the three parameters you used to evaluate bee reproduction
- All over the text: please add (just for the first mentioning in the text) authority to all scientific names and provide the related family

Reviewer 2 ·

Basic reporting

Overall, the paper was well-written. However, I felt the authors could do a better job of fitting their work into the broader literature. For example, they start out discussing diversity-stability relationships in the Introduction and then never, in the Discussion, mention what their findings might mean for this relationship. The Discussion has a total of 3 citations and only two of those put their findings in context. Also, the Introduction had a lot digressions (distance from floral patches, trip duration, etc.) that were not well explained and did not seem to fit directly into this study.

The figure should label the X and Y axis. The X axis is labelled but I would prefer simply stating the categories instead of A, B, C. Also, the full SEM should be put in the main text of the paper as it illustrates the main hypothesis they are working from.

Experimental design

The experimental design was well-explained and the research questions were interesting and easy to follow. The research was conducted in an ethical manner. Further, it explained the study with enough details so as to be ensure repeatability.

Validity of the findings

I am concerned with replication in this study. It is my understanding that a sample size of 5-10 is needed for each estimated parameter (based on James Grace's 2006 book). However, there are only 14 sites and for 5 parameters. This is obviously critical for this study as they are making the claim that resource abundance/richness/CV do not influence bee reproduction. I was surprised that none of these factors mattered. Also, I would like more explanation of what were the 'meta-analytic' methods used (L80-81).

Additional comments

This is an interesting study, that addresses important mechanisms underlying the relationship between floral abundance/richness/CV and bee reproduction. I liked the use of reproduction as a response variable as this is rarely done on so many species of bees in a single study. However. I was very surprised to see that none of the floral resource metrics influenced bee reproduction.

I would like to see how temporal stability affected bee reproduction when A) all species were combined. This would give one SEM model. I am wondering if some of the patterns are obscured by variation among species B) it would be useful to see how different species may or may not show patterns. One thing I am curious about is whether some bees that have longer phenological periods may show different patterns from those with shorter phenologies. For a bee whose flight time may be 2-3 weeks, the measure of resources (stability, richness, abundance etc.) that are not within their flight period will not likely be relevant. Resource availability needs to be assessed for each species.

INTRODUCTION

L39: no need for comma after ‘function’

L40: What are the many definitions of stability? Why is that sentence here? I found it diverted from the flow. Yes there are many definitions but it would be good to explain which is being used and why here.

L38-L50: Overall there are too many points in this one paragraph. I feel the authors are missing a chance to really highlight what is unique about their study: linking stability in resources to population metrics in a consumer. Everything else seemed to take away from this very cool point.

L46-48: Not sure what crop pollination has to do with this article.

L62-72: I had a difficult time understanding why a high diversity of flowers would lead to shorter foraging bouts. Also in the Gathmman and Tscharntke study, floral abundance as well as a diversity were correlated (as noted in their Discussion) so it is difficult to determine which (abundance vs. diversity) was really there driver.

L83-84: Another good paper here is: Rundlöf et al. 2014. Late-season mass-flowering red clover increases bumble bee queen and male densities. Biological conservation 172: 138-145

L86: I think the use of SEM is useful here. However, there could be a much better set-up for this in the introduction. This paragraph could be written more strongly to highlight this.

METHODS

L114-115: The selection of sites with ‘contrasting flower abundance, composition and diversity’ is an important point. However, there needs to be a better explanation of what contrasting means. How did they select these? Also, what was the mean/variance among sites for the parameters? It would be useful to know the variance that was in these parameters to get a better understanding as to whether the variation was large enough to find effects.

L126: A sentence explaining why they authors identified pollen would be useful.

L140: Change ‘he’ to ‘we’. Also, just state that you used flower density. The fact that you have some estimate of pollen amount per flower is digressive. As a reader, I now really want to know what that measure is and why you didn’t use it. Just delete that sentence. Flower density is standard and fine to use - don’t confuse the issue.


L155: I am somewhat confused about ‘bee fitness’ measures. Are these measures linked to an individual bee? The L98-99 all list per site measures. I realize it may not be possible to measure fitness of individual/phenotype/genotype but aren’t the measures in this study more appropriately defined as site-level reproduction?

L164: It would be good to have supplementary information on rarefied richness. Also the sentence in L165 seemed repetitive.

L165: The time since fire part needs to be explained earlier and not introduced in the stats section.

L166: a brief sentence explaining why standardization was done would be helpful.

L167: What was the package used for the SEM analysis?

L183: This is a new idea (generalization vs. specialization). It was only introduced in an offhand way on L70-72.

L188: Why do the authors use number of interactions as it is dependent on number of observations and thus highly problematic for determining specialization? Why not use a less biased metric for specialization such as d’?

L196-200: What package was used for the SEM? Also, it seems as if power is a bit low here. My understanding of SEM, is that an N of 5-10 replicates are needed for each estimated parameter. However, this SEM has 5 parameters and only 14 sites. Might this be a reason by there is no relationship among resources and reproduction?

DISCUSSION:

In general, the discussion needs to be put in the context of the literature much better. There are only 3 citations. How might their findings relate to the broader literature on diversity-stability relationships (as they start out with in their Intro)? They have a nice discussion about parasitism but this doesn’t relate back to anything in the Introduction.

L228-229: Did the authors test for direct effects of abundance/richness on bee reproduction? The full model only has indirect effects through stability. I find this result surprising.

L238-241: There needs to be more explanation as to why parasitism would change in the face of temporal stability.

L264: Why do the authors think specialization was not related to stability? This paragraph simply restates the results.

Figures: It would be helpful to simply label the X-axis with actual terms (total brood cells, mean number of brood cells, etc). Also, label the X-axis.

·

Basic reporting

The paper is generally well-written, although there are a few grammatical errors and some misleading citations. As examples: Wild bees are not gaining increased attention due to the results of Garibaldi et al. 2013. Rather that study is part of the increased attention that wild bees have been receiving, in large part due to perceived declines in honey bees. Blaauw & Isaacs (2014) studied sites where wildflowers were planted next to blueberries not the inverse. This is an important, if subtle distinction.

It is or should be common practice to cite taxonomic authorities with species names. These are used in Table 1., but Xylocopa splendidula was described by Lepeletier not Lepertier. In an English manuscript, it should be Toro and Cabezas, don't use 'y'. Osmia rufa is no longer in use. The correct name should be Osmia bicornis (Linnaeus).

Under trap nest sampling, it states that fig. S1 shows 6 points where traps were set up. I only see plant sampling locations in fig. S1.

Under plant sampling methods, it is stated that weekly sampling was considered appropriate because flowers last less than a week. Is this correct? Wouldn't that mean more frequent sampling would be necessary?

Citations should be checked carefully. For example, Minckley is incorrectly spelled in the biblography and Dieunomia and Helianthus in the same reference should be italicized. Other subtle flaws in the citations are also evident on quick inspection, Pat Willmer's middle initial may indeed by 'G.", but it is not used in the author list for Potts et al. 2003a, 2003b.

Experimental design

The experimental design seems appropriate, although one potential factor not included in models could be relevant. The authors measure floral diversity in 100 x 200 m sites, and include data on fire and elevation, but there does not seem to be an accounting of the surrounding habitat. Bees are capable of flying large distances, especially moderate to large bodied bees like the ones measured in this study. Some basic data on landscape at an appropriate radius around the sites (e.g., 0.5 - 2 km) would be a valuable factor to include in models. Some information on the biology of the bees would be useful since floral preference, constancy and adult phenology would be relevant to how they respond to floral stability and diversity. Do all bee species respond in the same way? Can you use different bee species are response variables to test this? And what about total productivity? The authors exclude species with fewer than 30 nests, but from a conservation stand point rare bees might be of interest. Do these different habitats support different numbers of bee species? It would be interesting to explore some of these ideas, even if they produce negative results.

Validity of the findings

The findings and their interpretation seem mostly reasonable in the context of the manuscript. I do think that there is probably too much speculation regarding the negative effect of temporal stability. Wasn't this non-significant? If the negative effect is non-significant does it require explanation?

---

## Round 0.2 · accepted · Accept

The manuscript has been carefully revised in agreement with reviewers' comments and can be accepted now.

·

Basic reporting

No comment

Experimental design

No comment

Validity of the findings

No comment

Additional comments

Dear colleagues,
thank you for having performed all my corrections.
Greetings